# Infrared Spectroscopy of RNA Nucleosides in a Wide Range of Temperatures

**DOI:** 10.3390/life14040436

**Published:** 2024-03-25

**Authors:** Susana Iglesias-Groth, Franco Cataldo, Martina Marin-Dobrincic

**Affiliations:** 1Instituto de Astrofisica de Canarias, Via Lactea s/n, 38200 La Laguna, Spain; 2Actinium Chemical Research Institute, Via Casilina 1626A, 00133 Rome, Italy; franco.cataldo@fastwebnet.it; 3Applied Physics and Naval Technology Department, Universidad Politécnica de Cartagena, C/Doctor Fleming, s/n., 30202 Cartagena, Spain; martina.marin@edu.upct.es

**Keywords:** astrobiology, astrochemistry, spectral line, identification, methods, laboratory, molecular, molecular data

## Abstract

The RNA world hypothesis suggests that early cellular ancestors relied solely on RNA molecules for both genetic information storage and cellular functions. RNA, composed of four nucleosides—adenosine, guanosine, cytidine, and uridine—forms the basis of this theory. These nucleosides consist of purine nucleobases, adenine and guanine, and pyrimidine nucleobases, cytosine and uracil, bonded to ribose sugar. Notably, carbonaceous chondrite meteorites have revealed the presence of these bases and sugar, hinting at the potential existence of nucleosides in space. This study aims to present the infrared spectra of four RNA nucleosides commonly found in terrestrial biochemistry, facilitating their detection in space, especially in astrobiological and astrochemical contexts. Laboratory measurements involved obtaining mid- and far-IR spectra at three temperatures (−180 °C, room temperature, and +180 °C), followed by calculating molar extinction coefficients (*ε*) and integrated molar absorptivities (*ψ*) for corresponding bands. These spectral data, along with *ε* and *ψ* values, serve to provide quantitative insights into the presence and relative abundance of nucleosides in space and aid in their detection.

## 1. Introduction

Nucleosides represent a step ahead in molecular complexity with respect to the nucleobases. When the purine and pyrimidine nucleobases are linked through the β-glycosidic bond to ribose or deoxyribose sugars, the resulting RNA or DNA nucleosides are obtained [1]. Based on contemporary life theory, the fundamental principle revolves around an RNA molecule’s ability to self-replicate independently, disseminating essential life functions’ information, thereby driving the quest for its nucleosides in space. The present study is focusing on the infrared spectroscopy of the RNA nucleosides whose chemical structures are shown in Figure 1.

According to the RNA world theory, in the pre-history of the biochemical evolution of life, RNA has played a key role both as genetic information storage for early life stages but also as a biochemical catalyst [2]. In other words, there was a stage in the pre-history of terrestrial biochemistry where primitive ancestors of modern cells were based exclusively on RNA for the storage of genetic information but also for the normal functioning of the cell, including the RNA replication [3]. Only in a later stage of biochemical evolution was the RNA specialized in the actual roles of information transfer and transduction, while the phenotype passed under the control of proteins through an RNA-peptide world transition stage [2].

The prebiotic synthesis of nucleosides has encountered many more difficulties than the prebiotic synthesis of nucleobases, because of the higher molecular complexity of the former [1]. However, more recent studies discovered potential and compelling prebiotic chemical routes for the abiotic production of nucleosides [4]. The key point regards the fact that the purine and pyrimidine nucleobases can be formed (as can the amino acids) either under prebiotic conditions on the Earth [5,6] but also in conditions occurring in the interstellar medium [7], including chemical processes occurring in star-forming regions [5,8]. Indeed, all the purines and pyrimidines nucleobases were found in carbonaceous chondrites [9], also including some nucleobases not currently used by terrestrial biochemistry. Similarly, sugar ribose was also detected in primitive meteorites [10]. Thus, although the nucleosides were not yet detected in primitive meteorites, it is possible to affirm that the key molecules to produce nucleosides (nucleobases from one side and ribose sugar from the other side) are present in primitive meteorites, and the glycosylation reaction which leads to nucleosides can be induced by cosmic rays [11]. In addition, nucleosides can be produced without the assistance of cosmic rays, but rather through wet chemistry conditions [8,12]. The study of nucleoside radiolysis has demonstrated that adenosine and guanosine, which are purine-based, are more resistant to radiolysis compared to cytidine and uridine, which are pyrimidine-based. This explains why adenine and guanine, the parent purine nucleobases, were more readily detected and found in higher concentrations in carbonaceous chondrites than cytosine and uracil, the parent pyrimidine RNA nucleobases [13,14].

On these premises, it is possible that RNA nucleosides may be detectable in space as is the case with nucleobases. The previous work on infrared spectroscopy was indeed dedicated to the nucleobases [15], while the present work is focusing on the infrared spectroscopy of the RNA nucleosides within our endeavors to explore spectroscopically and molecules of astrobiological relevance, like, for instance, amino acids, potentially detectable in space [16,17]. References to the significance of mid-infrared and far-infrared spectroscopy in studying astrobiologically relevant compounds, as well as the latest telescopes, may be found in the sources [18,19].

## 2. Materials and Methods

### 2.1. Materials

All the RNA nucleosides, i.e., purine-based adenosine (Ado) and guanosine (Guo), as well as the pyrimidine-based cytidine (Cyd) and uridine (Urd) (see Figure 1 for the chemical structures), used in the present work were purchased from Aldrich-Merck (St. Louis, MO, USA-Darmstadt, Germany). The cesium iodide spectroscopic matrix, which was used to create the pellets, was also acquired from Aldrich-Merck.

### 2.2. Preparation of Laboratory Apparatus and Pellets for Measuring the Integrated Molar Absorptivity in the Mid- and Far-Infrared Regions

Previous works [17,20] provided detailed descriptions of the laboratory equipment used for low and high temperature spectra in the mid- and far-infrared. Moreover, the comprehensive protocol on the manufacture of the pellets used to calculate the molar extinction coefficient (*ε*) and the integrated molar absorptivity (*ψ*) of the nucleobases studied in the present work can be found in [15,16,17]. Cesium iodide was used as a glue to compact the mix of our sample into pellets in order to conduct FT-IR spectroscopy. In these conditions, the sample analyzed can be considered the solute and the solid CsI matrix as the solvent. Therefore, operating the pellet sample is analogous to handling a solid solution. In such conditions, the determination of Epsilon (ε, molar extinction coefficient) and Psi (*ψ*, integrated molar absorptivity) values does not require a calibration curve, but a single measurement at a high dilution is sufficient.

## 3. Results and Discussion

### 3.1. Mid-Infrared Emissions of Ado, Guo, Cyd, and Urd and Their Molar Extinction Coefficients and Integrated Molar Absorptivity

The molar extinction coefficient (or molar attenuation coefficient) can be expressed using the Lambert-Beer law as follows:*ε* = *A* (*bc*)^−1^(1)
i.e., for a specific wavenumber $\tilde{\nu }$ = *λ*^−1^ or (wavelength *λ*) in the infrared spectrum, the absorbance *A* is multiplied by the path length b of the same matrix (expressed in cm) and the concentration of the sample (expressed in mol·L^−1^)) to obtain the molar extinction coefficient *ε* (expressed in L· mol^−1^ · cm^−1^) [17]. The integrated molar absorptivity *ψ* is obtained by integrating the absorbance across the full absorption band, with the $\tilde{\nu }$ in wavenumber on the x-axes:(2)$$\psi =\int_{band}\;\varepsilon d\tilde{\nu }={\left(bc\right)}^{-1}\int_{band}Ad\tilde{\nu }$$

In an infrared spectrum where wavenumbers are plotted on the x-axis, the integrated absorptivity is measured in units of cm^−1^. If the path length, denoted as *b*, is also expressed in centimeters (cm), and the concentration is given in moles per cubic centimeter (mol·cm^−3^), then the dimension of *ψ* is in centimeters per mole (cm·mol^−1^). To convert this into the astrochemical practical value of km·mol^−1^, we can simply multiply by a factor of 10^−5^ [15,16,17].

Table 1, Table 2, Table 3 and Table 4 show the outcomes of our assessment of both *ε* and *ψ* values of the RNA nucleosides adenosine (Ado), guanosine (Guo), cytidine (Cyd), and uridine (Urd) which are the building blocks for the chemical structure of the informational polymer RNA. 

The data are extracted from the spectra illustrated in Figure 1, Figure 2, Figure 3 and Figure 4. Concerning the integrated molar absorptivity, Table 1, Table 2, Table 3 and Table 4 also provide information on the integration ranges within which the *ψ* value for each pertinent band was calculated.

The spectra of the RNA nucleosides are presented with wavelength (μm) on the x-axis, aligning with the format preferred by astrophysicists and consistent with prior studies [15,16,17]. It is important to note that both the determination of band positions and the calculation of integrated molar absorptivity and molar extinction coefficients were conducted using spectra with the x-axis in wavenumbers (cm^−1^) which are the preferred units among spectroscopists. 

As previously mentioned in sections covering amino acids [15,16,17], it is important to emphasize that the molar absorptivities reported were obtained under specific laboratory conditions and should be used with caution when calculating the abundance of nucleobases in space. Although they can provide information about the comparative frequencies of these species when observed in similar spectral regions and allow approximate estimations of abundance levels, accurate determinations will require measuring the absorptivity values of nucleosides under laboratory conditions that closely mimic those found in interstellar space.

### 3.2. Mid-Infrared Spectroscopy of the Purine Nucleosides: Ado and Guo

There are several works on the infrared spectroscopy on the purine nucleosides adenosine (Ado) and guanosine (Guo) (see Figure 1 for the chemical structures) mainly made with the purpose of bands assignment, base pairing, hydrogen bonding, and as a key step toward the understanding of the infrared spectra of the nucleotides, polynucleotides, and RNA. Without aiming to be comprehensive, we mention a selection of papers dealing with the spectroscopy of purine nucleosides [21,22,23,24,25,26,27,28,29]. The primary objective of this study is to provide the astrophysics and astrochemistry/astrobiology community with the mid-infrared spectra of purine nucleosides (Figure 1 and Figure 2, Table 1 and Table 2) as a means of potentially identifying these compounds in condensed form in space.

Furthermore, the molar extinction coefficients (*ε*) and the integrated molar absorptivity (*ψ*) values (Table 1 and Table 2) were determined for the first time as a tool for the potential quantitative estimation of the amount of (Ado) or (Guo), in case they could be identified in space. Thus, we will not deal with the details of band assignments, already done effectively by many authors, rather, we will focus on some relevant infrared bands which are susceptible to large shifts in the wide temperature range studied. Since the nucleosides are more sensitive to thermal decomposition than the nucleobases [15,30] and the amino acids [16,17], the maximum temperature reached in this study was limited to 180 °C.

In line with earlier research on the spectroscopy of amino acids that are significant to astrobiology, covering a wide range of temperatures [15,16,17], in Table 1, Table 2, Table 3 and Table 4, we have provided a comprehensive analysis of the shift in the mid-infrared band with respect to temperature according to the following: Δ*ν* = *ν*_(−180 °C)_ − *ν*_(+180 °C)_(3)

We can see that Δ*ν* denotes the overall displacement of the band, encompassing the complete range of temperatures under consideration. The infrared bands susceptible of large or significant band shift on these molecules is typically assigned to hydrogen bond interaction [15,16,17]. Being a weaker bond than normal covalent bonds, the hydrogen bond is much more sensitive to temperature changes [31]. Consequently, the IR spectral investigation in a wide range of temperatures facilitates identification of the infrared bands involved in hydrogen bonding [32]. 

In Table 1 (see also Figure 1) are reported the infrared band shifts observed in Ado. Of course, the band shifts observed in Ado do not correspond necessarily to the shifts previously measured in the free nucleobase adenine. The reason is certainly due to the presence of the D-ribose ring attached to the purine base which leads to supplementary hydrogen bonding phenomena. At 3.00 μm (3335 cm^−1^), assigned to the OH groups of ribose [22], Δ*ν* = −19 cm^−1^ and more pronounced shift Δ*ν* = −38 cm^−1^ can be observed at 3.16 μm (3167 cm^−1^), assigned to the stretching of the NH_2_ group of Ado [22]. Other important band shifts are observed at 5.20 μm (1923 cm^−1^), with Δ*ν* = 10 cm^−1^ due to combination band, then 6.00 μm (1667 cm^−1^) Δ*ν* = 10 cm^−1^ due to scissoring of the NH_2_ group of Ado, while 6.23 μm (1605 cm^−1^) Δ*ν* = 9 cm^−1^ and 6.63 μm (1508 cm^−1^) Δ*ν* = 9 cm^−1^ are both assigned to the in-plane adenine ring mode [25,26,27]. Other vibrational modes characterized by relatively high band shift and assigned to the ribose ring of the Ado are those at 7.39 μm (1353 cm^−1^) Δ*ν* = 8 cm^−1^, 9.33 μm (1072 cm^−1^) Δ*ν* = 8 cm^−1^, 13.02 μm (795 cm^−1^) Δ*ν* = 9 cm^−1^, and 16.86 μm (593 cm^−1^) Δ*ν* = 10 cm^−1^ [22,25,26,27]. As shown in Figure 1 and Table 1, the maximum temperature reached in the FT-IR spectra recording on Guo was limited to 180 °C, although the melting point of Guo is found at 234°–237 °C. The latter temperature was never reached to avoid any possible thermal decomposition of Guo.

The other purine nucleoside, guanosine (Guo), is more thermally labile than Ado, although its melting point is reported at 239 °C. Figure 2 shows that at 180 °C, the infrared spectrum of Guo is strongly altered, especially at higher frequencies. Consequently, as shown in Table 2, the band shift Δ*ν* from 2.5 μm (4000 cm^−1^) to 5.77 μm (1733 cm^−1^) was determined in the range from −180 °C to +50 °C. Despite these limitations, large band shifts with temperature were measured at 3.12 μm (3209 cm^−1^) Δ*ν* = −9 cm^−1^ and 3.65 μm (2737 cm^−1^) Δ*ν* = −27 cm^−1^ assigned, respectively, to the N-H stretching and to the C-H stretching [23]. Then, apart from the 6.22 μm (1609 cm^−1^) Δν = 12 cm^−1^ due to guanine in-plane mode and the mixed modes due to guanine and ribose overlap both at 7.02 μm (1425 cm^−1^) Δν = 22 cm^−1^ and at 11.35 μm (881 cm^−1^) Δ*ν* = 9 cm^−1^, all the other bands with significantly large band shift reported in Table 2 are assigned to ribose: 7.29 μm (1372 cm^−1^) Δ*ν* = 8 cm^−1^, 8.85 μm (1130 cm^−1^) Δ*ν* = 13 cm^−1^, 9.79 μm (1021 cm^−1^) Δ*ν* = 9 cm^−1^, 11.64 μm (859 cm^−1^) Δ*ν* = 9 cm^−1^, 14.03 μm (713 cm^−1^) Δ*ν* = −11 cm^−1^, and 21.98 μm (455 cm^−1^) Δ*ν* = −9 cm^−1^ [23]. Even the glycosidic bond linking the purine guanine with ribose is subjected to band shift with temperature 8.45 μm (1179 cm^−1^) Δ*ν* = 9 cm^−1^ [23].

### 3.3. The Emissions of the Pyrimidine Nucleosides: Cyd and Urd in Mid-IR

Figure 3 presents the FT-IR spectra of the pyrimidine nucleoside cytidine (Cyd), while Table 3 provides the infrared bands along with their respective *ε* and *ψ* values. Since the melting point of Cyd is found in the range between 210° and 220 °C, the maximum temperature reached for the FT-IR recording was limited to 180 °C to avoid any decomposition which may occur on melting.

As in the case of the purine nucleosides, the infrared spectra in a wide range of temperatures permits to identify the infrared bands directly involved in hydrogen bonding as those which undergo the largest shifts. For Cyd, these bands are located at 2.90 μm (3448 cm^−1^) Δ*ν* = −24 cm^−1^, due to ribose OH stretching [33,34,35]. Other ribose assigned infrared bands which are involved in hydrogen bonding are as follows: 6.08 μm (1646 cm^−1^) Δ*ν* = −8 cm^−1^ ribose ring bending mode, 7.10 μm (1409 cm^−1^) Δ*ν* = 9 cm^−1^, 8.33 μm (1200 cm^−1^) Δ*ν* = 11 cm^−1^, 8.80 μm (1136 cm^−1^) Δ*ν* = 10 cm^−1^, 12.25 μm (816 cm^−1^) Δ*ν* = 12 cm^−1^, 21.65 μm (462 cm^−1^) Δ*ν* = 29 cm^−1^, and 23.75 μm (421 cm^−1^) Δ*ν* = −9 cm^−1^ [23,34,35]. Of course, a series of vibrational modes due to the cytosine moiety of Cyd are also involved in the hydrogen bonding, starting from the bending of the amino group at 6.67 μm (1500 cm^−1^) Δ*ν* = 11 cm^−1^ and followed by a series of bending modes of the pyrimidine ring at 8.67 μm (1153 cm^−1^) Δ*ν* = 9 cm^−1^, 13.61 μm (735 cm^−1^) Δν = −9 cm^−1^, 13.99 μm (715 cm^−1^) Δ*ν* = 15 cm^−1^, 17.39 μm (575 cm^−1^) Δ*ν* = 10 cm^−1^, and 17.73 μm (564 cm^−1^) Δ*ν* = 21 cm^−1^. 

The FT-IR spectrum of the other RNA pyrimidine nucleoside, uridine (Urd) is shown in Figure 4, and the summary of ε and *ψ* parameters can be found in Table 4. Among the RNA nucleosides, the simplest band pattern is displayed by Urd. Since the melting point of Urd is in the range 165°–168 °C, the FT-IR spectra of Urd were recorded from −180 °C to +150 °C.

In the case of Urd, there are only six infrared bands that show a significantly large band shift with temperature and are hence involved in hydrogen bonding. The band at 3.28 μm (3051 cm^−1^) Δ*ν* = −9 cm^−1^ is due to the OH stretching of the ribose moiety [26,27,34,36]. Similarly, the band at 3.47 μm (2878 cm^−1^) Δ*ν* = 9 cm^−1^ is also due to the C-H stretching of the ribose ring, while the band at 11.39 μm (878 cm^−1^) Δ*ν* = −17 cm^−1^ occurs in the bending modes of ribose [36]. There are three other infrared bands with a large shift with temperature in Urd, assigned to the uracil moiety of Urd at 5.95 μm (1681 cm^−1^) Δν = −15 cm^−1^, 7.16 μm (1397 cm^−1^) Δν = 8 cm^−1^, and 12.03 μm (831 cm^−1^) Δν = 8 cm^−1^ [26,27,34,36].

## 4. Conclusions

Astronomers now have access to the reference spectra of the four building blocks of the RNA molecule, adenosine (Ado), guanosine (Guo), cytidine (Cyd), and uridine (Urd), containing the basic information for the initial building of life. The presented spectra can be used to identify nucleosides qualitatively and quantitatively in space. The mid-infrared spectra of the nucleosides were measured at three distinct temperatures: −180 °C, +50 °C, and +180 °C (unless otherwise indicated, refer to Figure 1, Figure 2, Figure 3 and Figure 4). Table 1, Table 2, Table 3 and Table 4 provide a summary of the extent to which the bands shift in function of temperature for all the primary absorption bands. The values for the integrated molar absorptivity (*ψ*) and the molar extinction coefficients (*ε*) of the primary infrared bands of the nucleosides have been calculated and are shown in Table 1, Table 2, Table 3 and Table 4. With the available data, it will be feasible to search for—and potentially identify—the nucleosides and use the *ψ* and *ε* obtained here to assess their local abundance.

## Data Availability

All laboratory data reported in this paper are fully available from the authors.

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
