# Peer review of "Infrared Spectroscopy of RNA Nucleosides in a Wide Range of Temperatures"

_life, 2024, doi:10.3390/life14040436_

Round 1

Reviewer 1 Report

Comments and Suggestions for Authors

In this manuscript, authors present the IR method for RNA components: adenosine, guanosine, cytidine and uridine spectra detection and peaks shifting under a wide temperature change: from -180 to +180 °C. The molar extinction coefficients and molar absorptivity are calculated based on the peak shifts. Here are my comments about this manuscript before it can be published.

Major:

1.     DNA and RNA has very small structure difference on ribose, about this OH peak, can authors discuss why pick RNA and detect RNA instead of DNA? Especially the application is detecting information for astrophysicists and astrochemists, why RNA? A part of discussion or introduction should be claimed.

2.     In real condition for astrophysicist’s applications, how will other molecules affect the detection? To me this is like a lab measurement of pure RNA structure shifting in certain temperature range, how will this be useful for what authors claimed?

3.     In Figure 2, Guanosine, why is the peak below 5um disappeared when temperature is higher than 180?

4.     Since authors talked about the peak shift, the chemical structure is worth to add in the figure and major functional group peaks are worth to label out, to make the figures more straight forward to read.

Minor:

1.     There are some writing mistakes that should be corrected before publishing. In abstract, the line 10-12 is repetitive when reading.

2.     Line 23, the key words, the misused colons.

3.     Line 85, incomplete sentence.

4.     The interval in figures is inconsistent, can make better.

Author Response

In this manuscript, authors present the IR method for RNA components: adenosine, guanosine, cytidine and uridine spectra detection and peaks shifting under a wide temperature change: from -180 to +180 °C. The molar extinction coefficients and molar absorptivity are calculated based on the peak shifts. Here are my comments about this manuscript before it can be published.

Major:

  1. DNA and RNA has very small structure difference on ribose, about this OH peak, can authors discuss why pick RNA and detect RNA instead of DNA? Especially the application is detecting information for astrophysicists and astrochemists, why RNA? A part of discussion or introduction should be claimed.

THANK YOU FOR YOUR OBSERVATION. PLEASE NOTE THAT AT PAGE 2 OF THE MANUSCRIPT AND LINES 48-58 IT IS CLEARLY EXPLAINED WHY RNA INSTEAD OF DNA. ACCORDING TO THE CURRENT SYNTHETIC THEORY THE RNA IS MUCH OLDER THAN DNA AND RNA HAS PLAYED A KEY ROLE IN THE EARLY STAGES OF LIFE MOLECULAR EVOLUTION. INDEED, IT IS TALKED ABOUT “RNA WORLD” TO INDICATE THE EARLY STAGES OF MOLECULAR EVOLUTION WHEN RNA WAS ACTING AS ENZYMES, PROTEINS AND DNA. IN ADDITION WE HAVE ALREADY ADDRESSED THE INTEREST OF SIMILAR WORK FOR THE NUCLEOBASES OF THE DNA IN OUR RECENT PAPER PUBLISHED IN MONTHLY NOTICES OF THE ROYAL ASTRONOMICAL SOCIETY, IGLESIAS_GROTH AND CATLADO 2023.

  1. In real condition for astrophysicist’s applications, how will other molecules affect the detection? To me this is like a lab measurement of pure RNA structure shifting in certain temperature range, how will this be useful for what authors claimed?

THE SCOPE OF THIS WORK IS THAT TO OFFER A SERIES OF REFERENCE SPECTRA TO ASTROFISICISTS AND ASTROCHEMISTS (A&A). USING THESE SPECTRA THE A&A CAN EVENTUALLY IDENTIFY THESE MOLECULES IN SPACE AND USING THE MOLAR EXTINCTION COEFFICIENTS AND/OR INTEGRATED MOLAR ABSORPTIVITY, MAKE AN ESTIMATE OF THEIR ABUNDANCE IN SPACE, IF AND WHEN THEY WILL BE DETECTED. THE NEED TO RECORD THE SPECTRA AT DIFFERENT TEMPERATURE IS BECAUSE THE POSITION OF CERTAIN INFRARED ABSORPTION BANDS ARE SENSITIVE TO TEMPERATURE AND UNDERGO A SHIFT. IN PRINCIPLE, FROM THE BAND POSITION OF A TEMPERATURE SENSITIVE BAND IT COULD BE POSSIBLE TO ESTIMATE THE TEMPERATURE OF THE GIVEN COMPOUND. FURTHERMORE, AT LOWER TEMPERATURES THE BAND RESOLUTION IS LARGELY IMPROVED AND PERMIT TO RESOLVE BANDS THAT AT ROOM TEMPERATURE OR HIGHER TEMPERATURE APPEAR COALESCED INTO A SINGLE BROADER BAND….

  1. In Figure 2, Guanosine, why is the peak below 5um disappeared when temperature is higher than 180?

AS EXPAINED IN LINES 240-255 OF THE MANUSCRIPT, GUANOSINE IS ONE OF THE MOST SENSITIVE TO THERMAL DEGRADATION AMONG THE NUCLEOSIDES CONSIDERED HERE. AT 180°C SOME INCIPIENT DEGRADATION IS OBSERVED ONLY BELOW 5 MICONS.

  1. Since authors talked about the peak shift, the chemical structure is worth to add in the figure and major functional group peaks are worth to label out, to make the figures more straightforward to read.

YES, THIS IS A NICE IDEA, BUT IT IS TOO DIFFICULT TO PUT IN PRACTICE FOR A NUMBER OF REASONS SO THAT WE PREFER TO LEAVE THE DRAWINGS AS THEY ARE.

Minor:

  1. There are some writing mistakes that should be corrected before publishing. In abstract,

         the line 10-12 is repetitive when reading.

CORRECTED, THANKS

  1. Line 23, the key words, the misused colons.

CORRECTED, THANKS

  1. Line 85, incomplete sentence.

CORRECTED, THANKS

  1. The interval in figures is inconsistent, can make better.

YES, THE GAP BETWEEN FIG 1 AND FIG. 2 WAS NOW FILLED. FURTHERMORE, THE FIGURES ARE ACCOMPANIED BY TABLES OF THE ABSORPTION BANDS. THUS, a READER SHOULD CONCENTRATE ON THE TABLES WHEN REASONING ON BAND POSITION.

Reviewer 2 Report

Comments and Suggestions for Authors

A few questions and recommendations for the authors of the manuscript:

When analyzing samples obtained from meteorites, they usually work with a mixture of bases: adenosine, guanosine, and other nucleosides or bases. Many components will be present in one sample. Does your method identify the components of nucleoside/base mixtures? Are there characteristic areas in the IR spectra that allow us to determine the content of each nucleoside/base in the mixture?

Have the spectra of heterocyclic bases without ribose been recorded at the same temperature? Do the IR spectra of nucleosides and corresponding bases differ?  

It is advisable to superimpose the IR spectrum of guanosine on the IR spectrum of adenosine/cytidine and show different regions (at the same temperature). It is also desirable to apply the spectrum of cytidine and uridine.  

Where are IR spectroscopy experiments supposed to be conducted: on earth or in space? 

Line 46: This is not Scheme 1, but the Figure 1.

Lines 133, 138, 144, 150: There is no need to write the name of the nucleoside in capital letters in the table titles.  

Different size structures in Figure 1a.

Author Response

When analyzing samples obtained from meteorites, they usually work with a mixture of bases: adenosine, guanosine, and other nucleosides or bases. Many components will be present in one sample. Does your method identify the components of nucleoside/base mixtures? Are there characteristic areas in the IR spectra that allow us to determine the content of each nucleoside/base in the mixture?

THIS IS AN EXCELLENT QUESTION THAT PROBABLY IS LINKED TO THE UNDERSTANDING OF THE PURPOSE OF THIS WORK. FIRST OF ALL, IT IS NECESSARY TO CLARIFY THAT THE CONCENTRATION OF THE PURINE AND PYRIMIDINE NUCLEOBASES IN METEORITES IS EXTREMELY SMALL AND CAN BE MEASURED IN PPB (PARTS PER BILLION) OR PPM (PARTS PER MILLION) AT BEST…THE FT-IR SPECTROSCOPY IS NOT THE SUITABLE TECHNIQUE FOR THE DETECTION OF SUCH SMALL AMOUNTS OF COMPOUNDS AND IN FACT, THE PURINE AND PYRIMIDINE BASES ARE DETECTED THROUGH GAS-CHROMATOGRAPHY COUPLED WITH MASS SPECTROMETRY OR, BETTER WITH LIQUID CHROMATOGRAPHY COUPLED WITH MASS SPECTROMETRY.

THUS, OUR METHOD IS INTENDED AS A DATABASE WHERE THE FINGERPRINT OF EACH NUCLEOSIDE IN THE INFRARED IS RECORDED AT DIFFERENT TEMPERATURES AND THE INTENSITY OF THE BANDS ARE DETERMINED AS WELL. THIS DATABASE IS INTENDED AS A SERIES OF REFERENCE SPECTRA THAT THE ASTROPHYSICISTS AND ASTROCHEMISTS MAY USE AS REFERENCE WHEN THEY NEED TO IDENTIFY A MOLECULAR BAND PATTERN IN CERTAIN ASTROPHYSICAL OBJECT. ONCE THE GIVEN MOLECULE HAS BEEN IDENTIFIED FROM THE BAND PATTERN COMPARISON, THEN THROUGH THE MOLAR EXTINCTION COEFFICIENT OF THE MAIN ABSORPTION BANDS OR THROUGH THE INTEGRATED MOLAR ABSORPTIVITY IT WILL BE POSSIBLE TO IDENTIFY LINES WITH APPROPRIATE RELATIVE STRENGTH.

MAY BE THE MOST INSTRUCTIVE EXAMPLE OF SUCH PROCEDURE IS THAT USED BY Cami, J., Bernard-Salas, J., Peeters, E., & Malek, S. E. (2010). Detection of C60 and C70 in a young planetary nebula. Science, 329(5996), 1180-1182. THEY USED EXACTLY THIS PROCEDURE TO IDENTIFY THE FULLERENES IN THE YOUNG PLANETARY NEBULA TC-1.

Have the spectra of heterocyclic bases without ribose been recorded at the same temperature? Do the IR spectra of nucleosides and corresponding bases differ?  

YES, THE WORK WAS PUBLISHED SEE à Iglesias-Groth, S., & Cataldo, F. (2023). Mid-and far-infrared spectroscopy of nucleobases: molar extinction coefficients, integrated molar absorptivity, and temperature dependence of the main bands. Monthly Notices of the Royal Astronomical Society, 523(2), 1756-1771. THE HETEROCYCLIC BASES WITHOUT RIBOSE ARE CALLED NUCLEOBASES. OF COURSE THERE ARE SIGNIFICANT DIFFERENCES AMONG THE SPECTRA OF THE NUCLEOBASES AND THE NUCLEOSIDES.

It is advisable to superimpose the IR spectrum of guanosine on the IR spectrum of adenosine/cytidine and show different regions (at the same temperature). It is also desirable to apply the spectrum of cytidine and uridine.  

THANK YOU FOR THIS SUGGESTION. HOWEVER, WE PREFER TO AVOID TO IMPLEMENT IT SINCE THIS PAPER HAS ALREADY TOO MANY FIGURES AND TABLES AND ADDING MORE FIGURES WILL NOT IMPROVE THE CLARITY OF THIS WORK.

Where are IR spectroscopy experiments supposed to be conducted: on earth or in space? 

THE GOAL IS THAT THE REPORTED SPECTRA BE USED TO GUIDE THE DESIGN OF FUTURE SPECTROSCOPIC OBSERVATIONS USING SPACE TELESCOPES. FOR INSTANCE THE JAMES WEBB SPACE TELESCOPE; THE LARGEST SPACE OBSERVATORY IS EQUIPPED WITH EXCELLENT INFRARED SPECTROGRAPHS AS MIIR WHICH CAN PERFORM THE SPECTROSCOPY NEEDED TO IDENTIFICATION OF NUCLEOBASES.  AN EXAMPLE OF THIS KIND OF OBSERVATIONS WITH SPITZER ANOTHER SPACE OBSERVATORY CAN BE FOUND IN Cami et al. (2010). Detection of C60 and C70 in a young planetary nebula. Science, 329(5996), 1180-1182 WHERE YOU CAN FIND HOW THIS METHODOLOGY HAS BEEN SUCCESSFULLY APPLIED.

Line 46: This is not Scheme 1, but the Figure 1.

. THE TERM “SCHEME” IS DEDICATED TO THE PLOT OF THE CHEMICAL STRUCTURES WHILE THE TERM “FIGURES” IS DEDICATED TO THE INFRARED SPECTRA. WE THINK THE TERMINOLOGY WAS CORRECT

Lines 133, 138, 144, 150: There is no need to write the name of the nucleoside in capital letters in the table titles.  

THE NUCLEOSIDE IS WRITTEN IN CAPITAL LETTER TO PERMIT THE READER A RAPID IDENTIFICATION OF THE TABLE OF HIS/HER INTEREST.

Different size structures in Figure 1a.

FIGURE 1a = SCHEME 1, WE DO NOT AGREE, THE STRUCTURES HAVE THE SAME SIZE.

Round 2

Reviewer 1 Report

Comments and Suggestions for Authors

Since authors explain and corrected the comments, I don't see any reason from me to hesitate publishing this work. 
